# A practical preparation of bicyclic boronates via metal-free heteroatom-directed alkenyl $sp^2$-C–H borylation

Pei-Ying Peng[1,2], Gui-Shan Zhang[1,2], Mei-Ling Gong[1,2], Jian-Wei Zhang[1,2], Xi-Liang Liu[1,2], Dingding Gao [1,2], Guo-Qiang Lin [1,2], Qing-Hua Li [1,2✉] & Ping Tian [1,2✉]

Bicyclic boronates play critical roles in the discovery of functional materials and antibacterial agents, especially against deadly bacterial pathogens. Their practical and convenient preparation is in high demand but with great challenge. Herein, we report an efficient strategy for the preparation of bicyclic boronates through metal-free heteroatom-directed alkenyl $sp^2$-C–H borylation. This synthetic approach exhibits good functional group compatibility, and the corresponding boronates bearing halides, aryls, acyclic and cyclic frameworks are obtained with high yields (43 examples, up to 95% yield). Furthermore, a gram-scale experiment is conducted, and downstream transformations of the bicyclic boronates are pursued to afford natural products, drug scaffolds, and chiral hemiboronic acid catalysts.

[1] The Research Center of Chiral Drugs, Shanghai Frontiers Science Center for TCM Chemical Biology, Innovation Research Institute of Traditional Chinese Medicine, Shanghai University of Traditional Chinese Medicine, 1200 Cailun Road, Shanghai 201203, China. [2] China-Thailand Joint Research Institute of Natural Medicine, Shanghai University of Traditional Chinese Medicine, 1200 Cailun Road, Shanghai 201203, China. ✉email: qinghuali@shutcm.edu.cn; tianping@shutcm.edu.cn

Boronic acids[1] have been recognized as powerful substances in molecular recognition[2,3], chemical biology[4,5], materials science[6], and catalysis[7] in the past decades. Among them, the bicyclic boronates have attracted remarkable attention and research interests for organic and medicinal chemists owing to their unique structural features and properties. Such bicyclic skeletons exist not only in bioactive molecules (for example, taniborbactam[8], VNRX-7145[9], QPX7728[10], benzazaboroine-2[11], and naphthoxaborin-1[12]), but also in fluorescent sensors (such as naphthoxaborin-2[13]) and functional materials (for instance, benzazaboroine-1[14] and BN2VN[15]) (Fig. 1a)[16,17]. Particularly, the three compounds under clinical development as β-lactamase inhibitors, including taniborbactam, VNRX-7145, and QPX7728, are regarded as the present hope against deadly bacterial pathogens[18], and their scalable synthesis is in urgent need to supply the clinical demands[19]. Because of the growing importance of bicyclic boronates, substantial efforts have been devoted to their preparations. Recently, boron insertion emerged as a strategy for the synthesis of boron-containing products. In 2016 and 2017, Yorimitsu and coworkers successfully achieved nickel- and manganese-catalyzed boron insertion into the $sp^2$-C−O bond of benzofurans for the construction of bicyclic boronates (Fig. 1b)[13,20]. Subsequently, they developed transition-metal-free ring-opening borylation of indoles, in which boron atom was

inserted into the C2 − N bond using a large excess amount of lithium metal, to furnish 1,2-benzazaborins[21]. Very recently, Dong, Liu, and coworkers realized the boron insertion into the challenging $sp^3$-C − O bond in alkyl ethers through zinc/nickel tandem catalysis to provide bicyclic boronates (Fig. 1b)[22]. In addition, the palladium-catalyzed boron-selective biaryl coupling was also an efficient synthetic approach to versatile dibenzoxaborins[23]. It is worth mentioning that all these methods were based on the catalysis of transition metals (Ni, Zn, Mn, or Pd) or the use of excessive alkali metal (Li), which might limit their applications, particularly in consideration of industrial-scale production and the need to remove trace metals in active pharmaceutical ingredients[24].

Compared with the transition-metal-catalyzed process, the alternative metal-free strategy is usually practical, cost-effective, and environmentally benign. Early studies involving metal-free approach were reported by Dewar[25], Paetzold[26], and their co-workers, in which unprotected anilines or phenols could be converted to borazaroarenes and boroxaroarenes, respectively (Fig. 1c). However, their transformation suffered from harsh reaction conditions and the narrow substrate range[26].

With the recent development of metal-free C–H borylation of (hetero)arenes using boron reagents, such as $BX_3$ and $BH_3$[27,28], we became interested in the preparation of cyclic boronates from

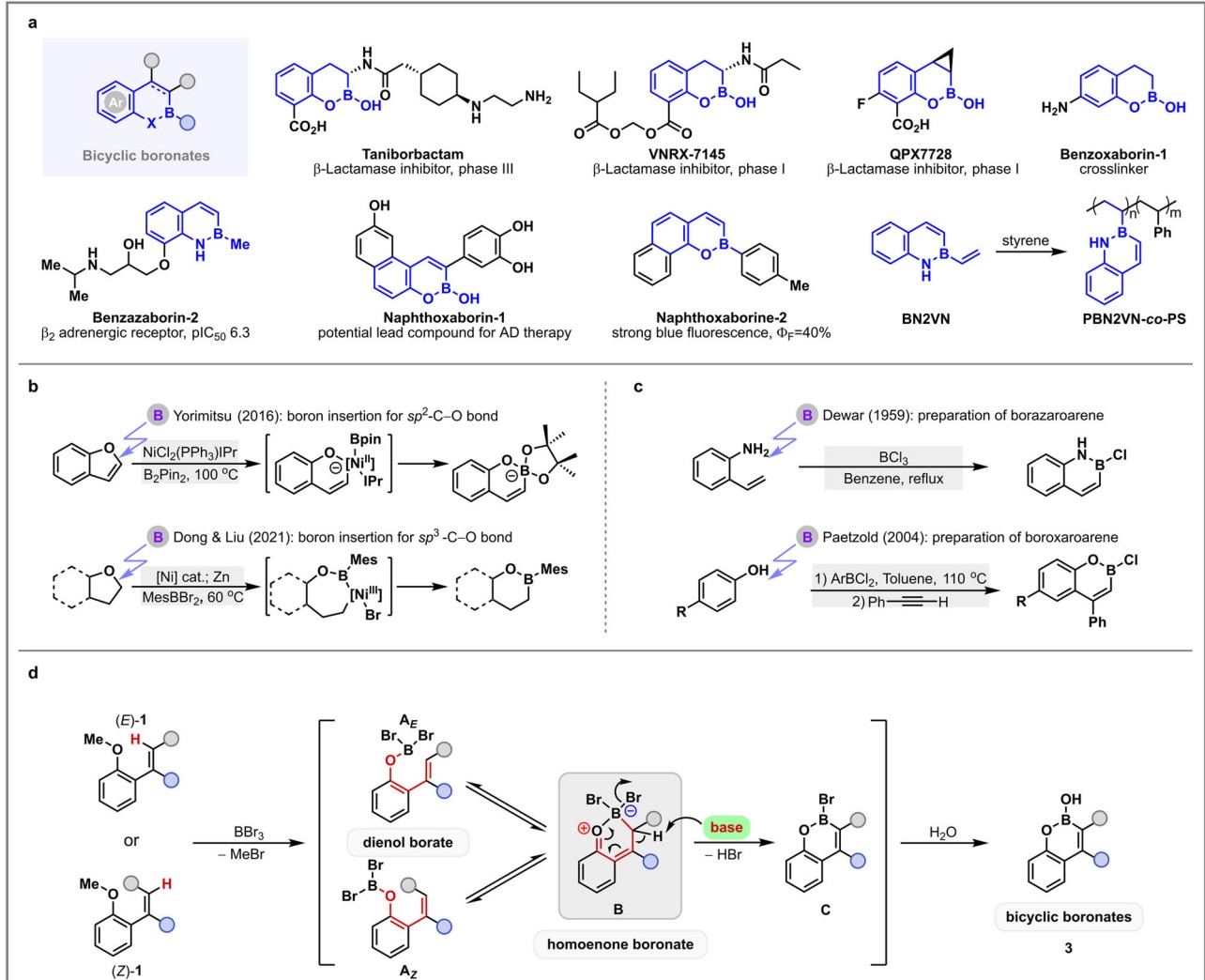

**Fig. 1 Strategy design for heteroatom-directed alkenyl $sp^2$-C–H borylation. a** Bicyclic boronate framework existing in drug molecules and functional materials. **b** Transition-metal-catalyzed boron insertion. **c** Metal-free approach to bicyclic boronates. **d** Metal-free O-directed alkenyl $sp^2$-C–H borylation.

alkenes via metal-free $sp^2$-C–H borylation[29,30]. To the best of our knowledge, the related research is quite rare in literature[24,31], probably due to the fact that alkenes can undergo cationic polymerization or haloboration under Lewis acidic conditions[31]. Inspired by the concept of "frustrated Lewis pairs", we propose that a suitable bulky base, just like a proton sponge, might be compatible with the borylation of alkenes without decreasing the activity of Lewis acid. Herein, our initial strategy is to utilize the oxygen atom in 2-(E)- or (Z)-vinyl anisole 1 as a directing group for $sp^2$-C–H borylation of neighboring olefin (Fig. 1d). Upon treatment of 1 with $BBr_3$, the corresponding borate $A_E$ or $A_Z$ is produced after the release of methyl bromide. Subsequently, the dienol borate $A_E$ or $A_Z$ can readily undergo a unique tautomerization through 1,5-sigmatropic rearrangement[32], in which 1,5-boron-shift occurred to give homoenone boronate B with a newly formed C–B bond. Compared with the direct intermolecular electrophilic borylation of terminal alkenes[29], our process takes advantage of intramolecular rearrangement of 2-vinylphenoxyborate ($A_E$ or $A_Z$) to gain the key ortho-quinone methide intermediate (B)[33,34], and such C–B bond construction can potentially improve the reaction efficiency and exclude the formation of different isomers. Due to the olefin double bond migration during the tautomerization process, both (Z)- and (E)- isomers of substrate 1 form a common homoenone boronate B. Next, under the promotion of a base, the elimination of HBr from intermediate B occurs, and the resulting intermediate C can be quenched by water to produce the desired bicyclic boronate 3. Obviously, the key in our strategy is to select an appropriate base to capture HBr. Because strong base tends to form a tight pair with $BBr_3$ and consequently prevents the cleavage of methyl ether bond, we focused on a range of bulky organic bases to promote this borylation process. As a matter of fact, such a special (Z)-alkenyl hemiboronic acid[35] is rarely documented, and its straightforward synthetic access is in great demand[36,37]. Herein, we report a practical preparation of bicyclic boronates via metal-free heteroatom-directed alkenyl $sp^2$-C–H borylation.

## Results and discussion

**Optimization of reaction conditions**. Our investigation started with the model reaction of o-isopropenylanisole (1aa), which is a component of the volatile oil of *Lippia javanica* and shows antimicrobial activities[38], and the selected results were summarized in Table 1. Initially, the reaction was carried out with commercially available borylation reagents, such as $BBr_3$, $BF_3$·$OEt_2$, and $BCl_3$ (2.0 equiv, 2a–c) in dichloromethane (DCM) at –60 °C for 0.5 h, however, only $BBr_3$ (2a) produced the desired bicyclic boronate 3aa in 35% yield, albeit with quantitative conversion (entry 1). As we envisioned, an appropriate base may suppress the occurrence of side reactions including cationic polymerization, hydrobromination, and bromoboration of alkene under acidic conditions. Thus some representative aliphatic and heterocyclic aromatic amines, such as N,N-diisopropylethylamine (DIPEA, **A1**)[30], pyridine (**A2**), and 2,6-lutidine (**A3**)[29], were selected for our reaction (entries 2–4). Both **A1** and **A2** resulted in low reaction conversion (entries 2 and 3). To our delight, the steric base, 2,6-lutidine **A3** led to 100% conversion and afforded **3aa** in 38% yield, indicating that Lewis acid $BBr_3$ could be compatible with bulky heterocyclic aromatic base. Several 2,6-disubstituted pyridines, for instance, 2,6-bis(trifluoromethyl)pyridine (**A4**), 2,6-di-*tert*-butylpyridine (**A5**, DTBP), and 2,6-diphenylpyridine (**A6**), were subsequently tested in our reaction. Among them, 2,6-di-*tert*-butylpyridine (**A5**) proved to be the most effective, affording **3aa** in almost quantitative yield (entry 6). The use of diphenyl pyridine (**A6**) considerably reduced the reaction conversion and yield (entry 7), and 2,6-bis(trifluoromethyl)pyridine (**A4**), which was less basic due to the

existence of two -$CF_3$ substitutions, surprisingly improved the yield of **3aa** from 38% to 89%. Another less basic base 2,3,5,6-tetramethylpyrazine (**A7**) was employed, and the desired bicyclic boronate **3aa** was achieved with 98% yield, which was quite comparable with the result from **A5** (entry 5). Next, the amounts of $BBr_3$ and **A5** were investigated. When a proportional excess of **A5** (2.0–1.1 equiv) relative to $BBr_3$ (1.5 equiv) was used, the yield of **3aa** dramatically decreased and significant amount of uncyclized side product o-isopropenyl phenol **1bc** was observed (entries 11–13). Interestingly, maintaining the 1:1 ratio of $BBr_3$ and **A5** allowed for further reduction in employing both reagents to 1.1 equiv. without an erosion of yield (entry 12). Lastly, raising or lowering the reaction temperature caused more side-reactions or decreased the reaction conversion, respectively (entries 13 and 14). As a result, the best yield for alkenyl $sp^2$-C–H borylation was achieved when $BBr_3$ (2a, 1.1 equiv) and 2,6-di-*tert*-butylpyridine (**A5**, 1.1 equiv) were used in DCM at –60 °C.

**Substrate scope of O-directed borylation of terminal alkenes**. With the optimal reaction conditions in hand, the scope of substrates was investigated, and the results are summarized in Fig. 2. First, different $R^1$ substituents at alkenyl moiety, such as methyl, ethyl, 2-ethyl methanesulfonate, and phenyl, were evaluated. All reactions proceeded smoothly in moderate to high yields (Fig. 2, **3aa–3ad**), and the structure of **3aa** was unambiguously confirmed by X-ray crystallographic analysis (CCDC 2143877 (**3aa**) contain the supplementary crystallographic data for this paper. These data can be obtained free of charge from The Cambridge Crystallographic Data Centre via www.ccdc.cam.ac.uk/data_request/cif.) (Supplementary Note 1 and Data 2). The substrate with $R^1$ = H could also afford the desired product **3ae**, albeit with slightly decreased yield of 78% (Fig. 2, **3ae**). Next, a variety of substituents $R^2$ at the phenyl ring were investigated, including alkyl (Me-, $^t$Bu-, $EtCO_2CH_2$-, or $CH_3(CH_2)_7NHCH_2$-), halogen (F-, Cl-, or Br-), trifluoromethyl and ether functionality (MeO- or 4-CN-$C_6H_4$O-). All the substitution groups were well tolerated, regardless of their location at C3-, C4-, C5-, or C6-position and their electron-withdrawing or electron-donating properties, in this borylative cyclization, with unambiguous confirmation of **3ah** structure by X-ray crystallographic analysis (CCDC 2143875 (**3ah**) contain the supplementary crystallographic data for this paper. These data can be obtained free of charge from The Cambridge Crystallographic Data Centre via www.ccdc.cam.ac.uk/data_request/cif.) (Fig. 2, **3af–3ax**, Supplementary Note 2 and Data 3). It is worth mentioning that methoxyl and N-pivaloyl group, which was reported for directed $sp^2$-C–H borylation of (hetero)arenes[24,27,39], did not affect our desired borylation process and only O-directed alkenyl boronic product was obtained in 92% yield (Fig. 2, **3ap**). Additionally, free -OH group at phenyl ring was also tolerated although excess base **A5** was required to inhibit side reactions, and the corresponding product **3ao** was obtained in 56% yield (Fig. 2, **3ao**). Naphthalene substrates also worked quite well to afford the desired boronates with moderate yields (Fig. 2, **3ay–3ba**). Notably, mono- and divinyl substituted chiral 2,2'-dimethoxy-1,1'-binaphthalene substrates **1az** and **1ba**, also proved to be suitable for this reaction, offering monohemiboronic acid (**3az**) and bishemiboronic acid (**3ba**) with excellent yields, respectively (Fig. 2, **3az** and **3ba**). The borylation of substrate **1az'** bearing free OH group required excess $BBr_3$ for full conversion, and 2'-naphthol product **3az'** was obtained with 75% yield (Fig. 2, **3az'**). Two estrone derivatives were also subjected to our reaction conditions. Interestingly, the treatment of methyl ether substate **1bb** with excess $BBr_3$ and **A5** could deliver the desired boronate **3ba**, along with some demethylation side product **1bb'**, while the phenol substrate **1bb'** failed

**Table 1 Optimization of Reaction Conditions[a].**

| Entry | BX₃ (2, n equiv) | A (y equiv) | T (°C) | t (h) | Conversion (%) | Yield(%)[b] |
|---|---|---|---|---|---|---|
| 1 | BBr₃ (2a, 2.0) | none | -60 | 0.5 | 100 | 35 |
| 2 | BF₃ (2b, 2.0) | none | -60 | 0.5 | 0 | / |
| 3 | BCl₃ (2c, 2.0) | none | -60 | 0.5 | 100 | trace |
| 4 | BBr₃ (2a, 2.0) | A1 (2.0) | -60 | 1 | <5% | trace |
| 5 | BBr₃ (2a, 2.0) | A2 (2.0) | -60 | 1 | <5% | trace |
| 6 | BBr₃ (2a, 2.0) | A3 (2.0) | -60 | 1 | 100 | 38 |
| 7 | BBr₃ (2a, 2.0) | A4 (2.0) | -60 | 1 | 100 | 86 |
| 8 | BBr₃ (2a, 2.0) | A5 (2.0) | -60 | 1 | 100 | 99 |
| 9 | BBr₃ (2a, 2.0) | A6 (2.0) | -60 | 1 | 70 | 60 |
| 10 | BBr₃ (2a, 2.0) | A7 (2.0) | -60 | 1 | 100 | 98 |
| 11 | BBr₃ (2a, 1.5) | A5 (2.0) | -60 | 2 | 100 | 50[c] |
| 12 | BBr₃ (2a, 1.5) | A5 (1.5) | -60 | 2 | 100 | 99 |
| 13 | BBr₃ (2a, 1.5) | A5 (1.1) | -60 | 2 | 100 | 99 |
| 14 | BBr₃ (2a, 1.1) | A5 (1.1) | -60 | 2 | 100 | 99 |
| 15 | BBr₃ (2a, 1.1) | A5 (1.1) | -40 | 2 | 100 | 86 |
| 16 | BBr₃ (2a, 1.1) | A5 (1.1) | -80 | 2 | 60 | 60 |

[a]Reactions were carried out with **1aa** (0.2 mmol, 1.0 equiv), BX₃ (**2**, n equiv), base (**A**, y equiv) in CH₂Cl₂ (1.0 mL) under N₂ atmosphere, unless otherwise noted.
[b]Determined by ¹H NMR analysis with mesitylene as an internal standard.
[c]o-isopropenyl phenol (**1bb**) was also obtained in 40% yield.

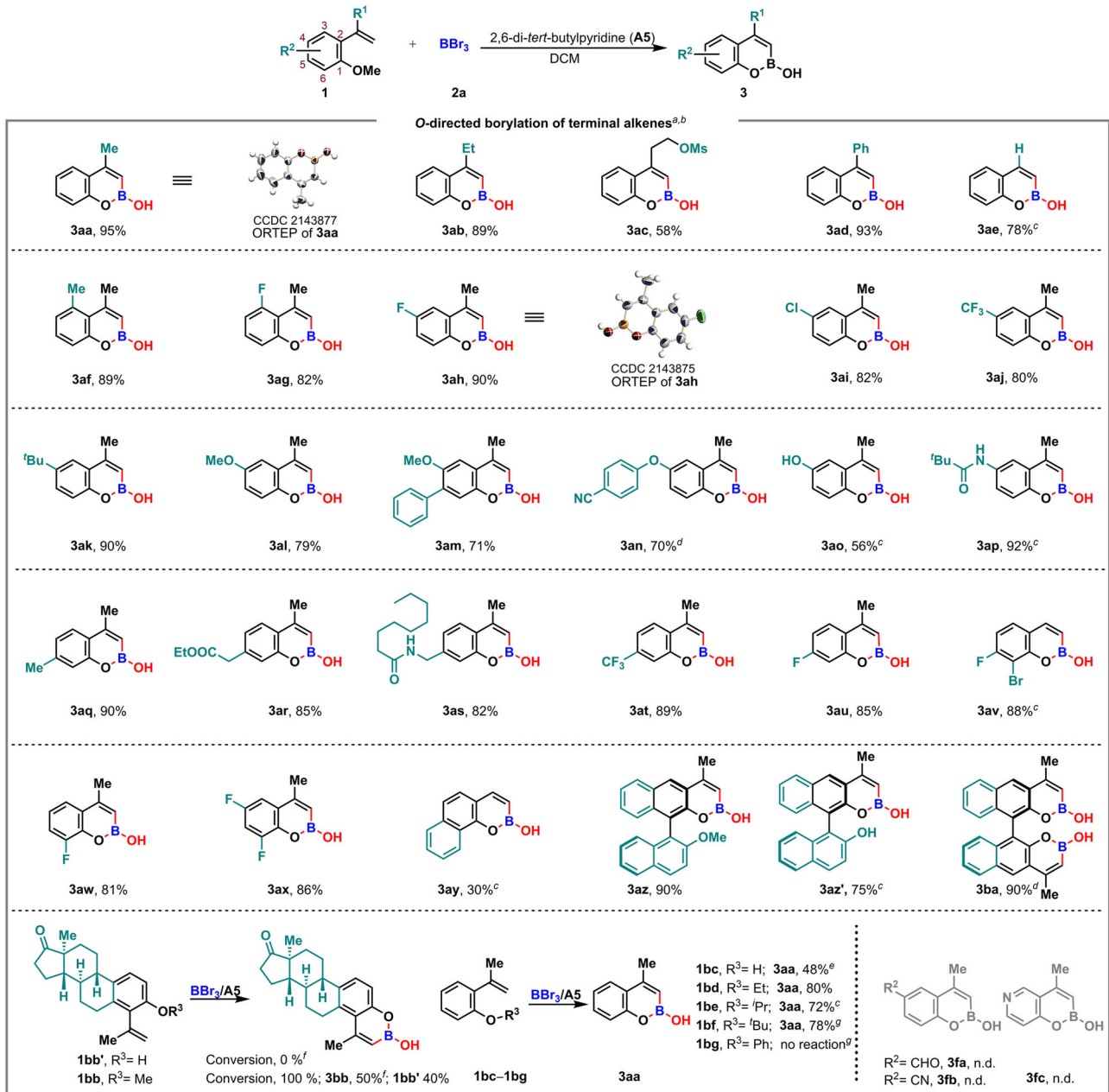

**Fig. 2 Reaction scope of *O*-directed borylation of terminal alkenes.** [a]Reactions were performed with **1** (0.2 mmol), **2a** (1.1 equiv, 1.0 M in DCM), **A5** (1.1 equiv) in DCM (1.0 mL) under N₂ atmosphere, −60 °C. [b]Yield of isolated product. [c]**2a** (3.0 equiv, 1.0 M in DCM), **A5** (1.1 equiv), −60 °C. [d]**2a** (2.0 equiv, 1.0 M in DCM), **A5** (2.0 equiv), −60 °C. [e]**A5** (2.0 equiv), −40 °C, 12 h. [f] **2a** (10.0 equiv, 1.0 M in DCM), **A5** (10.0 equiv), 0 °C, 8 h. [g]**2a** (2.0 equiv, 1.0 M in DCM), **A5** (2.0 equiv), 0 °C.

to undertake the borylation. At last, various *O*-linked groups R³, such as H, ethyl (Et), *iso*-propyl (*i*Pr), *t*-butyl (*t*Bu), benzyl (Bn), and phenyl (Ph), were evaluated. All substrates, except *O*-phenyl ether **1bg**, proceeded smoothly to give the desired product **3aa** with acceptable to moderate yields. Coincidently, the yields for two reactions with substrates **1bc**, **1ao**, and **1az'** were eroded by the existence of free -OH group, probably due to the consumption of **A5** by initially released HBr. It is worth mentioning that when the substituent R² is an electron-withdrawing group such as -CHO (**3fa**), -CN (**3fb**), or a substrate with a heteroaromatic ring (**3fc**), no desired product was detected in our standard conditions.

**Substrate scope of *N*- and *S*-directed borylation of terminal alkenes.** Upon achieving the above exciting results, we turned our

attention to other heteroatoms as a directing group. First, nitrogen atom was evaluated for this borylation reaction. Fortunately, the secondary aromatic amines worked very well as directing group to give desired products with moderate to good yields (Fig. 3, **3bh–3bk**). To our surprise, even the NH of heteroarenes, such as indole[24,40] and carbazole, could be employed as directing group to promote alkenyl *sp²*-C–H borylation with 91% and 72% yields, respectively (Fig. 3, **3bl** and **3bm**). Notably, the tertiary aromatic amine could also serve as directing group, affording a boronic acid product **3bn** in 52% yield (Fig. 3, **3bn**). Next, sulfur atom was introduced as directing group for our reaction, and the corresponding alkenyl *sp²*-C–H borylation was realized to give (*Z*)-alkenyl boronic acid product **3bo** with 42% yield. In addition, the stereochemistry of **3bh**, **3bm**, and **3bo** was unambiguously established by X-ray crystallographic analysis (CCDC 2143812

**Fig. 3 Reaction scope of *N*- and *S*-directed borylation of terminal alkenes.** [a]Reactions were performed using **1** (0.2 mmol), **2a** (2.0 equiv, 1.0 M in DCM), **A5** (2.0 equiv) in DCM (1.0 mL) under $N_2$ atmosphere, 0 °C. [b]Yield of isolated product. [c]**2a** (1.1 equiv, 1.0 M in DCM), **A5** (1.1 equiv) in DCM (1.0 mL), 0 °C.

(methyl ester of **3bh**), 2144121 (**3bm**), and 2143882 (**3bo**) contain the supplementary crystallographic data for this paper. These data can be obtained free of charge from The Cambridge Crystallographic Data Centre via www.ccdc.cam.ac.uk/data_request/cif.) (Supplementary Note 3–5 and Data 4–6). It is also worth mentioning that no desired product can be detected when the substrate contains -NH₂ (**1fd**) or -SH (**1fe**) group under the standard conditions.

**Substrate scope of *O*-directed borylation of (*E*)- and (*Z*)-alkenes.** To further expand the reaction scope, the substrates were extended to internal alkenes. First, all *o*-(*E*)-vinylanisole substrates, regardless of different R¹ (hydrogen or alkyl group) and R² (methyl, *n*-hexyl, phenyl or even forming a ring with R¹) substitutions, could successfully afford the corresponding products in moderate to good yields (Fig. 4, **3bp**–**3bu**). Next, the (*Z*)-alkenyl substrates **1bv** and **1bw** were subjected to this borylation, as expected, the corresponding bicyclic boronates **3bv** and **3bw** were successfully obtained with 95% and 81% yields, respectively. When (*Z*)- and (*E*)- isomers of trisubstituted alkene **1bx** (R¹ and R² = *n*-butyl) were separately treated with BBr₃ and **A5**, the same cyclization product **3bx** was achieved with excellent yield (Fig. 4, **3bx**). This result suggested that a mixture of *Z/E* isomers could lead to one single product. As a matter of fact, the treatment of a (*Z*)- and (*E*)-alkenyl mixture **1by** (*E/Z* = 1.06/1.00) under the borylation conditions indeed afforded the desired product **3by** in 93% yield, and its structure was further confirmed by X-ray crystallography (CCDC 2196049 (**3by**) contain the supplementary crystallographic data for this paper. These data can be obtained free of charge from The Cambridge Crystallographic Data Centre via www.ccdc.cam.ac.uk/data_request/cif. crystal data.) (Fig. 4, **3by**, Supplementary Note 6 and Data 7).

**Gram-scale experiment and synthetic transformations.** To demonstrate the synthetic applicability of this method, a gram-scale experiment of **1aa** was carried out and the corresponding product **3aa** was isolated with 95% yield. Moreover, several subsequent transformations of this boronic acid **3aa** were conducted to illustrate its unique utilities (Fig. 5A). First, the metal-free coupling between boroxine **3aa** and tosyl hydrazone delivered the *E*-alkene **4**

in a good yield[41], and its stereochemistry was established by X-ray crystallographic analysis (CCDC 2143883 (**4**) contain the supplementary crystallographic data for this paper. These data can be obtained free of charge from The Cambridge Crystallographic Data Centre via www.ccdc.cam.ac.uk/data_request/cif.) (Supplementary Note 7 and Data 8) . Two Pd-catalyzed reactions of **3aa** were also pursued. The Suzuki-Miyaura coupling with 4-bromotoluene led to (*Z*)-2-(1-phenylprop-1-en-2-yl)phenol **5** with 78% yield, and the carbonylation with carbon monoxide afforded 4-methyl coumarin **7** in 98% yield[39]. Under the copper-catalyzed conditions, 3-methyl benzofuran **8** was furnished from **3aa** through C–O bond formation with 91% yield[42]. The boronic acid **3aa** was also subjected to reduction or oxidation process. Hydrogenation over Pd/C could provide functionalized bicyclic boronate **6** in 90% yield, and the oxidative cleavage of carbon-boron bond with $H_2O_2$ resulted in the formation of 3-methyl-2-coumaranone **9** in 92% yield, likely through enol/hemiacetal intermediates[15]. Similar oxidation of **3bh** could generate 1,3-dimethylindole **10** through enol/hemiaminal intermediates in 95% yield (Fig. 5A). Next, a close chiral analog **11** of Hall's hemiboronic acid catalyst[43], in which the terminal phenyl ring was replaced with a cyclohexyl moiety, was designed and synthesized. The application of our protocol could conveniently convert the substrate **1bz** to the chiral catalyst **11**, which could promote the desymmetrization of 2-phenyl-propan-1,3-diol to provide chiral alcohol (*S*)-13 in 38% yield and with 85% ee (Fig. 5B). Finally, the utility of our method was demonstrated in a streamlined synthesis of (±)-QPX7728, an ultrabroad-spectrum inhibitor of serine and metallo-β-lactamases (Fig. 5C)[10,18]. Starting from the commercially available 3-bromo-4-fluoro-2-methoxy benzaldehyde, a four-step sequence, including Wittig reaction (94% yield), boron-insertion (88% yield), Simmons–Smith cyclopropanation (67% yield), and halogen–lithium exchange with dry ice quenching process (75% yield), successfully afforded (±)-QPX7728 in 42% overall yield[44]. Compared with the previous synthesis[19,44], this as an efficient alternative method is allowing for further optimization in the scalable production of QPX7728.

**Mechanistic considerations.** To gain further information regarding the reaction mechanism, we carried out ¹¹B NMR experiments (Fig. 6a). Compared with the chemical shift (δ 39.02 ppm) of pure BBr₃ in $CD_2Cl_2$, the addition of 2,6-di-*tert*-

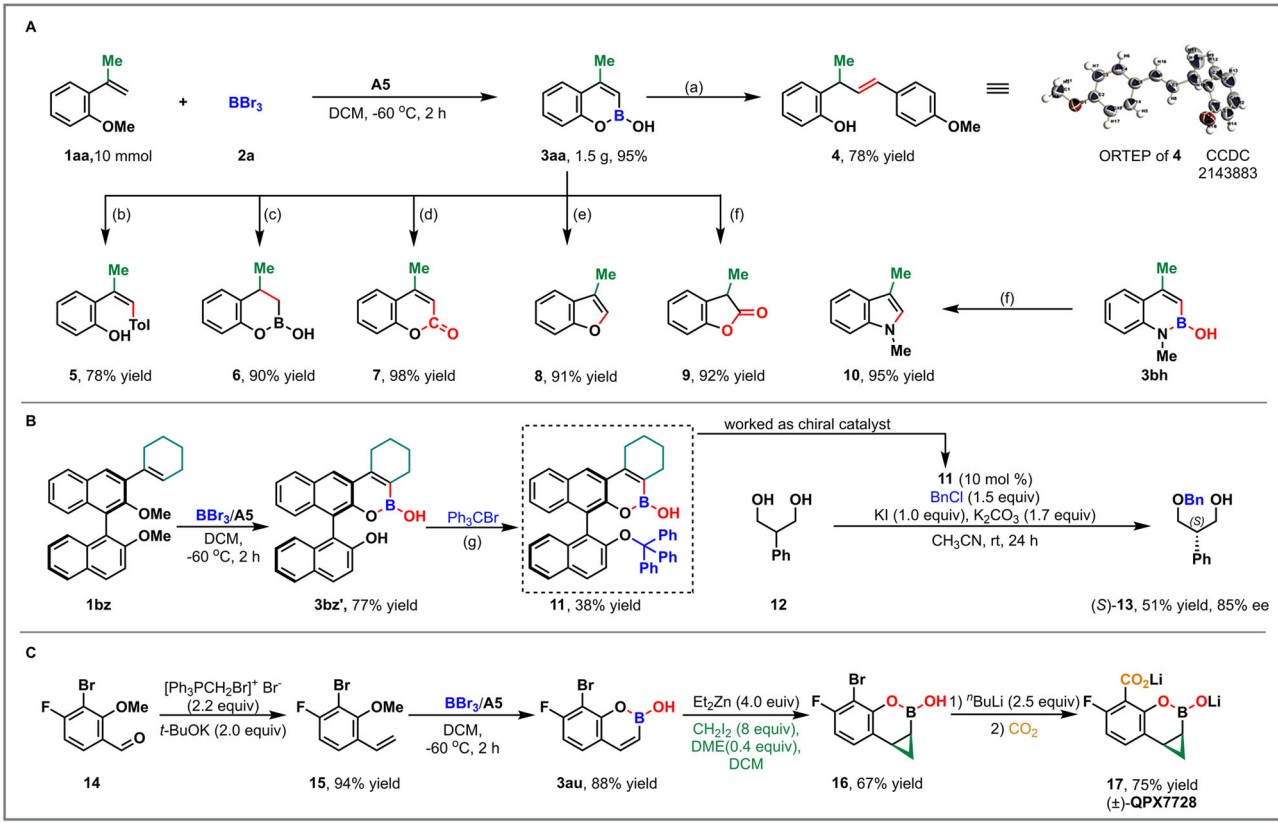

**Fig. 4 Reaction scope of O-directed borylation of (E)- and (Z)-alkenes.** [a]Reactions were performed using **1** (0.2 mmol), **2a** (1.1 equiv, 1.0 M in DCM), **A5** (1.1 equiv) in DCM (1.0 mL) under $N_2$ atmosphere, −60 °C. [b]Yield of isolated product. [c]−20 °C. [d]0 °C.

**Fig. 5 Synthetic applications. A**. Gram-scale experiment and several transformations: (a) *p*-Anisaldehyde tosylhydrazone (1.3 equiv), $K_2CO_3$ (1.3 equiv), dioxane (2.0 mL), 110 °C, 1.5 h; (b) 4-Bromotoluene (1.2 equiv), Pd(PPh_3)_4 (5 mol %), THF (2.0 mL), $K_2CO_3$ (2.0 equiv, 2.0 M aq), 80 °C, 5 h; (c) 10% Pd/ C (10 mol %), $H_2$ 6 atm, rt, 1.5 h; (d) Pd(OAc)_2 (1.0 equiv), CO, DMSO/MeOH (2.0 mL/1.0 mL), rt, 3.0 h; (e) Cu(OAc)_2 (0.2 equiv), Ag_2CO_3 (3.0 equiv), 1,10-phenanthroline (0.22 equiv), EtOH (2.0 mL), $H_2O$ (0.1 mL), in air, 80 °C, 22 h; (f) 30% $H_2O_2$ (1.0 mL), 3.0 N NaOH (1.0 mL), THF/EtOH (2.0 mL/ 0.5 mL), rt, 10 min. **B**. Synthesis and application of chiral hemiboronic acid catalyst: (g) Trityl bromide (1.0 equiv), Et_3N (1.0 equiv), DCM (2.0 mL), 60 °C, 4.0 h. **C**. Practical synthesis of (±)-QPX772.

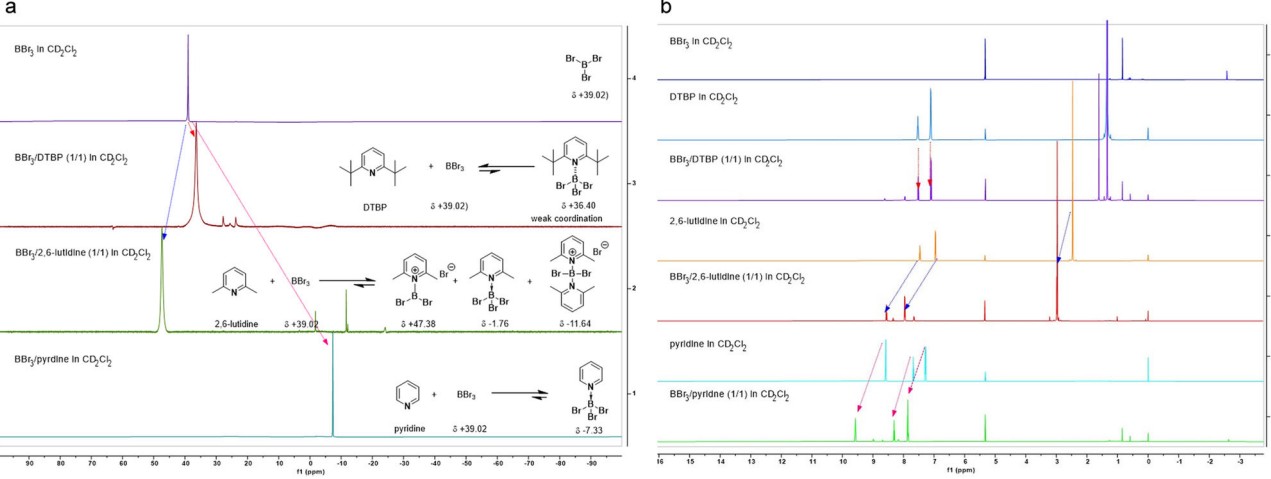

**Fig. 6 NMR experiments. a** $^{11}$B NMR stacked spectra. **b** $^1$H NMR stacked spectra.

butylpyridine (**A5**, 1 equiv) caused a slight high-field shift, giving only one signal (δ 36.40 ppm) in the positive chemical shift region. Interestingly, the addition of 2,6-lutidine (**A3**, 1 equiv) led to several $^{11}$B signals in both positive and negative chemical shift regions, and the major peak might be corresponding to a tight bromoborenium cation complex with 2,6-lutidine[45], causing the erosion of reaction yield. As we expected for a less hindered base, pyridine (**A2**) formed a stable complex with $BBr_3$, and the $^{11}$B NMR showed only one signal in the negative chemical shift region (δ −7.33 ppm). $^1$H NMR experiments also showed a similar trend for the coordination of $BBr_3$ with different bases. Upon the addition of 1 equivalent of $BBr_3$, significant downfield shifts were observed for both **A3** and **A2**, while little shifting effect was observed for **A5** (Fig. 6b). Therefore, both $^{11}$B and $^1$H NMR spectroscopic characterization strongly supported our initial strategy design, in which sterically bulky base could slightly decrease the Lewis acidity of $BBr_3$ through relatively weak coordination and mainly serve as an effective proton scavenger.

## Conclusion

In conclusion, we have developed a practical, concise, and convenient synthetic method for the construction of bicyclic boronates through metal-free heteroatom-directed alkenyl $sp^2$-C-H borylation. The reactions worked efficiently with good to excellent yields and were well compatible with various functional groups. More importantly, the alkene starting materials, in either (*Z*)- or (*E*)-form, could produce the desired bicyclic boronates, which significantly expanded the substrate scope of such borylative cyclization. Additionally, a gram-scale reaction for bicyclic boronates and several subsequent synthetic transformations were also presented to elaborate the valuable utilities. In particular, a practical synthesis of the ultrabroad-spectrum β-lactamase inhibitor (±)-QPX7728 was developed.

## Method

**General procedure for the preparation of product 3aa**. A dried Schlenk flask was charged with 2,6-di-*tert*-butylpyridine (**A5**, 0.22 mmol, 1.1 equiv) and 0.8 mL of DCM at −60 °C under nitrogen atmosphere. $BBr_3$ (**2a**, 0.22 mmol, 1.1 equiv; 1.0 M solution in DCM) was subsequently added dropwise while stirring. After the reaction mixture was stirred for 5 min, a solution of **1aa** (0.20 mmol, 1.0 equiv) was added dropwise and the resulting mixture was stirred for another 2.0 h at −60 °C. Finally, the reaction was quenched with 2,6-di-*tert*-butylpyridine (**A5**, 1.0 equiv), methanol (1.0 mL), and water (0.2 mL). Upon the removal of solvents in vacuo, the residue was purified by flash silica gel (300-400 mesh) chromatography (petroleum ether/ethyl acetate = 5/1) to afford the desired product **3aa** (30 mg) in 95% yield as white solid.

## Data availability

Detailed experimental procedures and characterization of compounds can be found in the Supplementary Information. The X-ray crystallographic structure reported in this study has been deposited at the Cambridge Crystallographic Data Centre (CCDC) under deposition numbers CCDC 2143877 (**3aa**), 2143875 (**3ah**), 2143812 (methyl ester of **3bh**), 2144121 (**3bm**), 2143882 (**3bo**), 2196049 (**3by**), and 2143883 (**4**). These data can be obtained free of charge from The CCDC via www.ccdc.cam.ac.uk/data_request/cif. All data are available from the authors upon request. NMR spectra as a separate Supplementary Data 1. All cif files as Supplementary Data 2–8.

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

## Acknowledgements

Financial support was generously provided by the National Key R&D Program of China (2022YFF1202600), the National Natural Science Foundation of China (22071155, 21871184, and 81903423), the Science and Technology Commission of Shanghai Municipality (21ZR1460700, 20XD1403600, 19YF1449300, and 20400750300), the Shanghai Municipal Education Commission (2019-01-07-00-10-E00072), and the Innovation Team and Talents Cultivation Program of National Administration of Traditional Chinese Medicine (No: ZYYCXTD-202004).

## Author contributions

P.-Y.P., G.-S.Z. and X.-L.L. performed the synthetic experiments and analysed the experimental data; M.-L.G. and D.G. performed the antibacterial experiments; J.-W.Z. performed the X-ray crystallographic analysis; G.-Q.L., Q.-H.L. and P.T. directed the research and prepared the manuscript. All the authors discussed the results and commented on the final manuscript.

## Competing interests

P.T., Q.-H. L., P.-Y.P., M.-L.G., G.-S.Z. and G.-Q.L. are inventors on CN patent application no. 2022104886919 filed by the Shanghai University of Traditional Chinese Medicine covering the preparation of bicyclic boronates and their applications in antibacterial agents. Other authors declare no competing interests.
