## [Peer Review File · Communications Chemistry]

Reviewers' comments:

Reviewer #1 (Remarks to the Author):

The authors report a metal-free heteroatom-directed alkenyl sp²-C-H borylation. The alkene starting materials, in either (Z)- or (E)-form, could produce the desired bicyclic boronates, which significantly expanded the substrate scope of such borylative cyclization. In general, this manuscript is well-written and clear, the science is interesting and I consider it to be high-impact based on the ability to generate bicyclic boronates. Also noteworthy is the demonstration of the downstream transformations of products to afford natural products, drug scaffolds, and chiral hemiboronic acid catalysts. The work should be of broad appeal, however, significant revisions and additional experiments are required before the work is suitable for publication in COMMSCHEM.

- 1) What's the reactivity of BF₃, BCl₃ and BI₃? The author should add these important information in the reaction optimization.
- 2) The author should test an example like 2,5-dimethoxy-4-(prop-1-en-2-yl)-1,1'-biphenyl to check which part (alkenyl or aryl) is prefer to cyclization.
- 3) Regarding the mechanism, the authors proposed the ortho-quinone methide intermediate. However, I didn't see a solid evidence to prove this step. The authors should try some gem-substituted alkenes. Could they observe the rearrangement product. Furthermore, the quinone may be stabilized.
- 4) The authors proposed the dienol borate formed ahead of C-B bond formation. Why OMe in 3al is compatible. Is it possible that BBr₃ coordinates with OMe and then activates the alkenyl group? They should give explanations and more experiments to support the proposed pathway
- 5) In Supporting information, starting material and final product was synthesized, purified and thoroughly characterized by NMR, IR and HRMS. Although the quality of the data is good, the text of the SI should be proofread and carefully checked. There are multiple typos, editing issues, etc. Furthermore, the NMR spectra of some compounds, e.g. 1bf, 1ar, 1ba, 1bf, 3ba, 3bh and 3bj, show that these compounds are not pure enough. These samples must be re-purified.

Reviewer #2 (Remarks to the Author):

This manuscript describes a novel and useful method for preparing bicyclic boronates using a [1,5]-sigmatropic rearrangement without metal catalysis and will be of high interest to readers in the field. It is clearly written and well-referenced; the characterization of compounds is thorough and the spectral data show high purity. Overall, the subject and quality warrant publication; however, there are certain notable absences in the demonstration of scope that should be addressed prior to acceptance. Specifically, missing from the various substituents on the aromatic ring that have been explored are those that are strongly electron-withdrawing by resonance, such as -CO₂R or -CN, particularly in the

positions ortho and para to the alkenyl group. The inductively electron-withdrawing substituent F does not sufficiently explore this question. In addition, for the N-directed and S-directed borylation of terminal alkenes (Fig. 3), results with directing groups NH₂ and SH are notable absences. These examples are needed to fully test the generality of the method and should be included.

In addition, there are a number of other items that should be addressed, as follows:

- In Figure 1d, it is recommended that structure B show a positive charge on O and a negative charge on B.
- The values in Table 1 do not appear to match the text. Specifically, the text says “When excessive A5 (2.0 equiv.) was used, the yield of 3aa dramatically decreased and significant amount of uncyclized side product o-isopropenylphenol 1bb was observed (entries 9–11).” However, entries 9-11 in Table 1 all show 100% conversion and 99% yield. Also, 2.0 equiv. of A5 is the condition used in Entry 6; the authors should make clear in describing the later examples that they are referring to excessive A5 relative to BBr₃.
- The text describing Fig. 5 appears to be corrupted, i.e. immediately after reference 42, a sentence begins with the uncapitalized word “functionalized” and does not make sense.
- The use of this methodology to synthesize the beta-lactamase inhibitor QPX7728 provides a nice demonstration of its utility. However, the implication that this represents a 4-step synthesis from a commercially-available starting material, and the statement “compared with the previous synthesis, our concise and practical approach was cost-effective” ignores the fact that the starting material 3-bromo-4-fluoro-2-methoxybenzaldehyde costs an exorbitant amount (\$500,000/kg based on gram amounts). The wording should be revised to simply describe this as an efficient alternative method; while it is important to acknowledge the published synthesis, cost comparisons should be avoided. Also, as noted above, it would be useful to know if the boron insertion can be achieved with the carboxylate moiety already in place (presumably as an ester).
- It is not at all clear how an MIC of 2.67 µg/mL against E. coli is obtained for compound 3bp. It is generally accepted that the MIC corresponds to “no growth.” According to the plot on page S31 of the Supporting Info, this is achieved at 2.2 log₁₀, which would be well over 100 µg/mL. If it is the case that all the MICs are ≥100 µg/mL, then the antibacterial evaluation should be removed entirely from the manuscript since many organic compounds have modest antibacterial activity, in most cases by non-specific and uninteresting mechanisms of action. On the other hand, if the MIC vs. E. coli is in fact <3 µg/mL, it is indeed an interesting result that is worthy of reporting.

Reviewer #3 (Remarks to the Author):

Tian and co-workers report a new, transition metal-free synthetic protocol for bicyclic boronate synthesis via heteroatom-directed C(sp²)-H borylation. These compounds are relatively rare in the literature but are found in several important β-lactamase inhibitors, as well as fluorescent sensors and functional materials. Hence, this report is likely to be of significant interest to the synthetic chemistry community, including medicinal chemists.

The scope of the method is fairly broad, and both electron-rich and electron-poor arene groups are compatible, as well as different alkene substitution patterns. However, no Lewis basic heteroaromatic

substrates (e.g., methoxy vinyl pyridines) are demonstrated, and one would assume that they would complex the BBr₃ non-productively? Notably, the reaction can be carried out cleanly on gram scale, which highlights its practicality.

The authors also provide extensive examples of further derivatisations of the bicyclic boronates, which is great to see. Moreover, they applied their method to the synthesis of the ultrabroad-spectrum β -lactamase inhibitor (\pm)-QPX7728, and also showed that several of the boronates displayed antibacterial activity against gram-negative bacterium *E. coli*.

This is a very well put together manuscript and was an enjoyable read. The Supporting Information is prepared to a good standard and the NMR spectra are all of high quality. Given the novelty of the approach to bicyclic boronates and the widespread interest in these unusual compounds, I have no hesitation in recommending this work for publication in Communications Chemistry.

General Response to Reviews

We appreciate the time and effort that the editor and reviewers dedicated to providing feedback on our previous manuscript (Ms. No **COMMSCHEM-22-0574A**) and are grateful for the insightful comments on and valuable improvements to our paper. We have carefully reviewed the comments and have revised the manuscript extensively. Our response is given in a point-to-point manner as follows.

Reviewer #1:

The authors report a metal-free heteroatom-directed alkenyl sp²-C-H borylation. The alkene starting materials, in either (Z)- or (E)-form, could produce the desired bicyclic boronates, which significantly expanded the substrate scope of such borylative cyclization. In general, this manuscript is well-written and clear, the science is interesting and I consider it to be high- impact based on the ability to generate bicyclic boronates. Also noteworthy is the demonstration of the downstream transformations of products to afford natural products, drug scaffolds, and chiral hemiboronic acid catalysts. The work should be of broad appeal, however, significant revisions and additional experiments are required before the work is suitable for publication in COMMSCHEM.

[Q1] 1) What's the reactivity of BF₃, BCl₃ and BI₃? The author should add these important information in the reaction optimization.

[Q2] 2) The author should test an example like 2,5-dimethoxy-4-(prop-1-en-2-yl)-1,1'-biphenyl to check which part (alkenyl or aryl) is prefer to cyclization.

[Q3] 3) Regarding the mechanism, the authors proposed the ortho-quinone methide intermediate. However, I didn't see a solid evidence to prove this step. The authors should try some gem-substituted alkenes. Could they observe the rearrangement product. Furthermore, the quinone may be stabilized.

[Q4] 4) The authors proposed the dienol borate formed ahead of C-B bond formation. Why OMe in 3al is compatible. Is it possible that BBr₃ coordinates with OMe and then activates the alkenyl group? They should give explanations and more experiments to support the proposed pathway.

[Q5] 5) In Supporting information, starting material and final product was synthesized, purified and thoroughly characterized by NMR, IR and HRMS. Although the quality of the data is good, the text of the SI should be proofread and carefully checked. There are multiple typos, editing issues, etc. Furthermore, the NMR spectra of some compounds, e.g. 1bf, 1ar, 1ba, 1bf, 3ba, 3bh and 3bj, show that these compounds are not pure enough. These samples must be re- purified.

Response to Reviewer 1

[Q1] 1) What's the reactivity of BF₃, BCl₃ and BI₃? The author should add these important information in the reaction optimization.

Response: Thanks for the reviewer's advice. We tried other commercially available

borylation reagents, such as $\text{BF}_3 \cdot \text{OEt}_2$ and BCl_3 , and the experimental results have been added in Table 1 (Optimization of Reaction Conditions). Because the reagent BI_3 was expensive and need very long delivery period, so we did not test it.

[Q2] 2) The author should test an example like 2,5-dimethoxy-4-(prop-1-en-2-yl)-1,1'-biphenyl to check which part (alkenyl or aryl) is prefer to cyclization.

Response: Thanks for the reviewer's advice. The substrate 2,5-dimethoxy-4-(prop-1-en-2-yl)-1,1'-biphenyl (**1am**) was synthesized and tested in our standard reaction condition to produce the desired product **3am** with 71% yield, while no aryl borylation product (**3am'**) was observed. In addition, the **3am** was added in Fig 2 as a new compound and the data of characterizations was added in SI (page S4, S16, S54 and S99).

[Q3] 3) Regarding the mechanism, the authors proposed the ortho-quinone methide intermediate. However, I didn't see a solid evidence to prove this step. The authors should try some gem-substituted alkenes. Could they observe the rearrangement product. Furthermore, the quinone may be stabilized.

Response: Thanks for the reviewer's advice. At the beginning of preparing the manuscript, we proposed an intermediate **Int-AB**, which is a resonance of **B**. After discussing with colleagues and reading related literature, we thought intermediate **B** was more stable, so it was presented in the manuscript. Very recently, Michael Ingleson and coworkers reported "Haloboration of *o*-Alkynyl Phenols Generates Halogenated Bicyclic-Boronates" and discussed a similar intermediate **I3A** which

is the structure of *ortho*-quinone methide.

Our work:

Michael Ingleson's work:

Angew. Chem. Int. Ed. **62**, e202301463 (2023)

Of course, according to reviewer's advice, a gem-substituted alkene **1ca** was synthesized and tested in our reaction condition. From crude ¹³C-NMR and ¹¹B-NMR, a peak of 185 ppm which indicated the formation of quinone methide was found. However, when we wanted to prepare the stable pinacol boronate (**3ca**), the signal of quinone disappeared and no desired product **3ca** was obtained. Also, we tried to grow a crystal, but compound **3a** cannot form a solid for long time under freezing condition.

[Q4] 4) The authors proposed the dienol borate formed ahead of C-B bond formation. Why OMe in **3al** is compatible. Is it possible that BBr_3 coordinates with OMe and then activates the alkenyl group? They should give explanations and more experiments to support the proposed pathway.

Response: Thanks for the reviewer's advice. Because both lone pair electrons and π electrons can coordinate with Lewis acid BB_3 , when there is only 1.1 equivalent of BBr_3 under our standard conditions, the complex of lone pair electrons and π electrons simultaneously coordinated with BB_3 is the major product and participates in the following reaction. So, this is maybe the reason of -OMe compatible in **3al** and **3am**.

In addition, we proposed that both routes were possible and could lead to intermediate of quinone methide. Because the first stage of path b is a reversible reaction, the possibility of the irreversible path a is higher.

[Q5] 5) In Supporting information, starting material and final product was synthesized, purified and thoroughly characterized by NMR, IR and HRMS. Although the quality of the data is good, the text of the SI should be proofread and carefully checked. There are multiple typos, editing issues, etc. Furthermore, the NMR spectra of some compounds, e.g. **1bf**, **1ar**, **1ba**, **1bf**, **3ba**, **3bh** and **3bj**, show that these compounds are not pure enough. These samples must be re-purified.

Response: Thanks for your kind suggestions! The SI have been carefully re-checked. The typo errors and editing issues have been corrected. In addition, the impure compounds, such as **1bf**, **1ar(1as)**, **1ba**, **1bf**, **3bh** and **3bj**, have been re-purified, and their yield and NMR spectra have also been revised in the modified manuscript and SI. For the compound **3ba**, we have repeatedly prepared it many times and the reported result is the best, also we have tried to re-purify it by preparative chromatography, but the impurity became more and more.

1-(*tert*-Butoxy)-2-(prop-1-en-2-yl)benzene (**1bf**)

Following the general procedure on 1.0 mmol scale, the substrate **1bf** was obtained as colorless oil. Eluent: PE, 143 mg, 75% yield; ¹H NMR (600 MHz, CDCl₃) δ 7.22 – 7.20 (m, 1H), 7.18 – 7.15 (m, 1H), 7.03 – 7.00 (m, 2H), 5.10 (s, 1H), 5.06 (s, 1H), 2.15 (s, 3H), 1.35 (s, 9H); ¹³C NMR (151

MHz, CDCl₃) δ 153.2, 145.8, 138.8, 129.6, 127.5, 123.2, 122.9, 115.1, 79.7, 29.1, 23.1; HRMS (ESI): Exact mass calcd for C₁₃H₁₉O[M+H]⁺: 191.1436, Found: 191.1432.

4-(*tert*-Butyl)-1-methoxy-2-(prop-1-en-2-yl)benzene (**1as(1ar)**)

Following the general procedure on 3.0 mmol scale, the substrate **1ar(1as)** was obtained as colorless oil. Eluent: PE/EA (10:1), 0.5 g, 52% yield; ¹H NMR (600 MHz, CDCl₃) δ 7.13 (d, *J* = 7.6 Hz, 1H), 6.81 (d, *J* = 7.5 Hz, 1H), 6.80 (s, 1H), 5.75 (s, 1H), 5.14 – 5.13 (m, 1H), 5.04 – 5.03 (m, 1H), 4.42 (d, *J* = 5.7 Hz, 2H), 3.82 (s, 3H), 2.27 – 2.17 (m, 2H), 2.09 (s, 3H), 1.68 – 1.62 (m, 2H), 1.35 – 1.23 (m, 10H), 0.87 (t, *J* = 7.0 Hz,

3H); ^{13}C NMR (151 MHz, CDCl_3) δ 172.9, 156.8, 143.8, 138.8, 132.0, 129.5, 119.7, 115.2, 110.4, 55.5, 43.5, 36.8, 31.8, 29.3, 29.3, 29.1, 25.8, 23.1, 22.6, 14.1; **HRMS (ESI)**: Exact mass calcd for $\text{C}_{20}\text{H}_{32}\text{NO}_2[\text{M}+\text{H}]^+$: 318.2433, Found: 318.2431.

(8*R*,9*S*,13*S*,14*S*)-2-Methoxy-13-methyl-1-(prop-1-en-2-yl)-6,7,8,9,11,12,13,14,15,16-decahydro-17*H*-cyclopenta[*a*]phenanthren-17-one (1ba(1bb))

Following the general procedure on 6.0 mmol scale, the substrate **1ba** was obtained as white solid; Mp: 213–215 °C; Eluent: PE/EA (10:1), 0.5 g, 26% yield; $^1\text{H NMR}$ (600 MHz, CDCl_3) δ 7.20 (d, $J = 8.6$ Hz, 1H), 6.76 (d, $J = 8.6$ Hz, 1H), 5.29 (s, 1H), 4.79 (s, 1H), 3.80 (s, 3H), 3.10–2.87 (m, 1H), 2.73–2.57 (m, 1H), 2.50 (dd, $J = 19.1$, 8.6 Hz, 1H), 2.44–2.38 (m, 1H), 2.29–2.23 (m, 1H), 2.19–2.09 (m, 1H), 2.09–2.04 (m, 1H), 2.00–1.95 (m, 5H), 1.66–1.46 (m, 6H), 1.42 (br, 1H), 0.91 (s, 3H). $^{13}\text{C NMR}$ (151 MHz, CDCl_3) δ 221.0, 154.4, 142.5, 134.7, 132.4, 132.0, 124.5, 115.1, 108.2, 55.8, 50.5, 48.0, 44.5, 37.9, 35.9, 31.6, 27.9, 26.8, 26.1, 23.3, 21.6, 13.8. **HRMS (ESI)**: Exact mass calcd for $\text{C}_{22}\text{H}_{28}\text{NaO}_2[\text{M}+\text{Na}]^+$: 347.1987, Found: 347.1984.

(3a*S*,3b*R*,11b*S*,13a*S*)-8-Hydroxy-10,13a-dimethyl-3,3a,3b,4,5,8,11b,12,13,13a-decahydrocyclopenta[7,8]phenanthro[3,2-*e*][1,2]oxaborinin-1(2*H*)-one (3bb(3ba))

Following the general procedure on 0.1 mmol scale, 2.0 mmol BBr₃ (10.0 equiv) and 2.0 mmol **A5** (10.0 equiv) were used, and the reaction mixture was stirred at 0 °C for 8.0 h. Eluent: PE/EA (5:1), light yellow solid, 17 mg, 50% yield; Mp: 231–233 °C; ¹H NMR (600 MHz, MeOD) δ 7.34 (d, *J* = 8.6 Hz, 1H), 7.07 (d, *J* = 8.6 Hz, 1H), 5.95 (s, 1H), 3.21 – 3.14 (m, 1H), 2.64 (s, 3H), 2.49 (m, 1H), 2.41 – 2.33 (m, 1H), 2.28 (dd, *J* = 15.8, 7.4 Hz, 1H), 2.18 – 2.09 (m, 1H), 2.09 – 1.97 (m, 2H), 1.92 – 1.86 (m, 1H), 1.73 – 1.62 (m, 1H), 1.62 – 1.44 (m, 4H), 1.34 – 1.25 (m, 2H), 0.93 (s, 3H). ¹³C NMR (151 MHz, MeOD) δ 198.2, 157.9, 152.1, 135.9, 134.7, 126.7, 124.1, 116.9, 50.3, 45.3, 37.4, 35.4, 31.6, 30.6, 29.2, 26.6, 26.3, 21.0, 13.0. ¹¹B NMR (193 MHz, Methanol-*d*₄) δ 26.60. **HRMS (ESI)**: Exact mass calcd for C₂₁H₂₆BO₃[M+H]⁺: 337.1975, Found: 337.1972.

1,4-Dimethylbenzo[e][1,2]azaborinin-2(1H)-ol (3bh)

Following the general procedure, the reaction was performed at 0 °C. Eluent: PE/EA (10:1 to 5:1), white solid, 22 mg, 63% yield; Mp: 92–95 °C; ¹H NMR (600 MHz, MeOD) δ 7.73 (dd, *J* = 7.9, 1.6 Hz, 1H), 7.44 – 7.42 (m, 1H), 7.35 (dd, *J* = 8.5, 1.2 Hz, 1H), 7.06 (t, *J* = 1.1 Hz, 1H), 6.35 (d, *J* = 1.2 Hz, 1H), 3.34 (s, 3H), 2.52 (d, *J* = 1.2 Hz, 3H); ¹³C

NMR (151 MHz, MeOD) δ 152.6, 143.4, 128.2, 125.6, 124.2, 118.5, 113.5, 28.7, 22.1;
 ^{11}B NMR (193 MHz, MeOD) δ 27.95; **HRMS (ESI)**: Exact mass calcd for $\text{C}_{10}\text{H}_{13}\text{BNO}[\text{M}+\text{H}]^+$: 174.1090, Found: 174.1081.

1-Cyclohexylmethyl -4-methylbenzo[e][1,2]azaborinin-2(1H)-ol (3bj)

Following the general procedure, the reaction was performed at 0 °C.

Eluent: PE/EA (10:1 to 5:1), white solid, 32 mg, 62% yield; Mp: 136–138

°C; ¹H NMR (600 MHz, CDCl₃) δ 7.79 (d, *J* = 8.0 Hz, 1H), 7.54 – 7.43

(m, 1H), 7.41 (d, *J* = 8.4 Hz, 1H), 7.13 (t, *J* = 7.6 Hz, 1H), 6.30 (s, 1H),

3.94 (d, *J* = 6.7 Hz, 2H), 2.53 (s, 3H), 1.89 – 1.85 (m, 1H), 1.75 – 1.54 (m, 5H), 1.20 – 1.08 (m,

5H); ¹³C NMR (151 MHz, CDCl₃) δ 152.1, 142.5, 128.0, 126.3, 125.5, 118.9, 115.2, 48.9, 36.5,

31.4, 26.7, 26.1, 23.2; ¹¹B NMR (193 MHz, MeOD) δ 26.65; HRMS (ESI): Exact mass calcd

for C₁₆H₂₃BNO[M+H]⁺: 256.1873, Found: 256.1872.

Reviewer #2:

This manuscript describes a novel and useful method for preparing bicyclic boronates using a [1,5]-sigmatropic rearrangement without metal catalysis and will be of high interest to readers in the field. It is clearly written and well-referenced; the characterization of compounds is thorough, and the spectral data show high purity. Overall, the subject and quality warrant publication; however, there are certain notable absences in the demonstration of scope that should be addressed prior to acceptance. [Q1] Specifically, missing from the various substituents on the aromatic ring that have been explored are those that are strongly electron-withdrawing by resonance, such as $-\text{CO}_2\text{R}$ or $-\text{CN}$, particularly in the positions ortho and para to the alkenyl group. The inductively electron-withdrawing substituent F does not sufficiently explore this question.

[Q2] In addition, for the N-directed and S-directed borylation of terminal alkenes (Fig. 3), results with directing groups NH_2 and SH are notable absences. These examples are needed to fully test the generality of the method and should be included.

In addition, there are a number of other items that should be addressed, as follows:

[Q3] In Figure 1d, it is recommended that structure B show a positive charge on O and a negative charge on B.

[Q4] The values in Table 1 do not appear to match the text. Specifically, the text says "When excessive AS (2.0 equiv.) was used, the yield of 3aa dramatically decreased and significant amount of uncyclized side product o isopropenylphenol 1bb was observed (entries 9-11)." However, entries 9-11 in Table 1 all show 100% conversion and 99% yield. Also, 2.0 equiv. of AS is the condition used in Entry 6; the authors should make clear in describing the later examples that they are referring to excessive AS relative to BBr_3 .

[Q5] The text describing Fig. S appears to be corrupted, i.e. immediately after reference 42, a sentence begins with the uncapitalized word "functionalized" and does not make sense.

[Q6] • The use of this methodology to synthesize the beta-lactamase inhibitor QPX7728 provides a nice demonstration of its utility. However, the implication that this represents a 4-step synthesis from a commercially-available starting material, and the statement "compared with the previous synthesis, our concise and practical approach was cost-effective" ignores the fact that the starting material 3-bromo-4-fluoro-2-methoxybenzaldehyde costs an exorbitant amount (\$\$00,000/kg based on gram amounts). The wording should be revised to simply describe this as an efficient alternative method; while it is important to acknowledge the published synthesis, cost comparisons should be avoided.

[Q7] Also, as noted above, it would be useful to know if the boron insertion can be achieved with the carboxylate moiety already in place (presumably as an ester).

[Q8] • It is not at all clear how an MIC of 2.67 $\mu\text{g}/\text{ml}$ against E.coli is obtained for compound 3bp. It is generally accepted that the MIC corresponds to "no growth." According to the plot on page S31 of the Supporting Info, this is achieved at 2.2 log₁₀, which would be well over 100 $\mu\text{g}/\text{ml}$. If it is the case that all the MICs are 100 $\mu\text{g}/\text{ml}$,

then the antibacterial evaluation should be removed entirely from the manuscript since many organic compounds have modest antibacterial activity, in most cases by non-specific and uninteresting mechanisms of action. On the other hand, if the MIC vs. *E. coli* is in fact $<3 \mu\text{g/ml}$, it is indeed an interesting result that is worthy of reporting.

Response to Reviewer 2

[Q1] Specifically, missing from the various substituents on the aromatic ring that have been explored are those that are strongly electron-withdrawing by resonance, such as $-\text{CO}_2\text{R}$ or $-\text{CN}$, particularly in the positions ortho and para to the alkenyl group. The inductively electron-withdrawing substituent F does not sufficiently explore this question.

Response: Thanks for the reviewer's advice. During this research, we have planned to introduce an ester group in the positions ortho and para to the alkenyl group and tried to develop a concise route to synthesize QPX7728-Na, however, when the strongly electron-withdrawing substituents, such as $-\text{CO}_2\text{R}$, $-\text{CHO}$, $-\text{CN}$, and $-\text{NO}_2$ was introduced to the aromatic ring, the substrate become inert under the standard condition (presented in the following scheme). If the **A5** was absent from the reaction system, only demethylated by-product was found without bicyclic boronate.

[Q2] In addition, for the N-directed and S-directed borylation of terminal alkenes (Fig. 3), results with directing groups NH_2 and SH are notable absences.

Response: Thanks for the reviewer's advice. The substrate **1ea** with amine was explored under the same reaction condition as the substrate **1bh**, although the

reaction time was extended to 12 h at room temperature, there was also no reaction.

The substrate **1fe** with the directing group SH was difficult to synthesize. We have tried some routes and finally obtained 50 mg through the following scheme E. We tried various reaction conditions, and changed the temperature, time, the amount of 2,6-di-tert-butylpyridine (**A5**), to our frustration, the results were that the **1fe** disappeared, but no desired product was observed. The reviewer can also find the reaction scheme and NMR spectra of starting material as follows.

O-(2-(prop-1-en-2-yl)phenyl) dimethylcarbamothioate (**1fc**)

$^1\text{H NMR}$ (600 MHz, CDCl_3) δ 7.30 – 7.26 (m, 2H), 7.24 – 7.21 (m, 1H), 7.05 (d, $J = 8.3$ Hz, 1H), 5.17 – 5.16 (m, 1H), 5.06 – 5.05 (m, 1H), 3.44 (s, 3H), 3.31 (s, 3H), 2.07 (s, 3H).; $^{13}\text{C NMR}$ (151 MHz, CDCl_3) δ 187.6, 150.6, 141.3, 136.7, 129.0, 127.6, 125.9, 124.0, 116.0, 43.2, 38.6, 23.7.; **HRMS (ESI)**: Exact mass calcd for $\text{C}_{12}\text{H}_{15}\text{NOS}[\text{M}+\text{Na}]^+$: 244.0767, Found: 244.0754.

S-(2-(prop-1-en-2-yl)phenyl) dimethylcarbamothioate (1fd)

¹H NMR (600 MHz, CDCl₃) δ 7.51 (dd, *J* = 7.8, 1.5 Hz, 1H), 7.33 (td, *J* = 7.5, 1.4 Hz, 1H), 7.26 (td, *J* = 7.6, 1.6 Hz, 1H), 7.22 (dd, *J* = 7.6, 1.7 Hz, 1H), 5.16 (p, *J* = 1.6 Hz, 1H), 4.88 (dd, *J* = 2.1, 1.0 Hz, 1H), 3.01 (d, *J* = 39.0 Hz, 6H), 2.09 – 2.03 (m, 3H).; ¹³C NMR (151 MHz, CDCl₃) δ 166.8, 148.7, 145.2, 137.4, 129.1, 128.5, 127.2,

126.0, 115.5, 36.7, 24.2; **HRMS (ESI)**: Exact mass calcd for C₁₂H₁₅NOS[M+H]⁺: 222.0947,
Found: 222.0956.

2-(prop-1-en-2-yl)benzenethiol (1fe)

$^1\text{H NMR}$ (600 MHz, CDCl_3) δ 7.60 (dd, $J = 7.8, 1.4$ Hz, 1H), 7.21–7.18 (m, 1H), 7.17–7.15 (m, 1H), 7.12–7.11 (m, 1H), 5.33–5.32 (m, 1H), 5.02–5.01 (m, 1H), 2.12–2.12 (m, 3H).; $^{13}\text{C NMR}$ (600 MHz, CDCl_3) δ 143.6, 142.9, 134.1, 128.1, 127.8, 126.7, 126.3, 117.4, 24.2. **HRMS (ESI)**: Exact mass calcd for $\text{C}_9\text{H}_{10}\text{S}[\text{M}]^+$: 150.0498, Found: 150.0503.

[Q3] • In Figure 1d, it is recommended that structure B show a positive charge on O and a negative charge on B. The reviewer can also find them

Response: Thanks for the reviewer's advice. The charge has been added on the O and B in the revised manuscript, and the reviewer can also find it as follows.

[Q4] • The values in Table 1 do not appear to match the text. Specifically, the text says "When excessive AS (2.0 equiv.) was used, the yield of 3aa dramatically decreased and significant amount of uncyclized side product o isopropenylphenol 1bb was observed (entries 9-11)." However, entries 9-11 in Table 1 all show 100% conversion and 99% yield. Also, 2.0 equiv. of AS is the condition used in Entry 6; the authors should make clear in describing the later examples that they are referring to excessive AS relative to BBr₃.

Response: Thanks for the reviewer's advice. The descriptions have been corrected in the revised manuscript, and the reviewer can also find it as follows.

~~Next, the amounts of BBr₃ and A5 were investigated. When excessive A5 (2.0 equiv.) was used, the yield of 3aa dramatically decreased and significant amount of uncyclized side product o isopropenyl phenol 1bb was observed (entries 9-11). When a proportional excess of A5 (2.0-1.1 equiv) relative to BBr₃ (1.5 equiv) was used, the yield of 3aa dramatically decreased and significant amount of uncyclized side product o isopropenyl phenol 1bc was observed (entries 11-13).~~ Interestingly, maintaining

[Q5] The text describing Fig. S appears to be corrupted, i.e. immediately after reference 42, a sentence begins with the uncapitalized word "functionalized" and does not make sense.

Response: We very much appreciate the reviewer's advice, the typo error has been corrected in the revised manuscript, and the reviewer can also find it as follows.

in 98% yield.⁴⁰ ~~F~~functionalized bicyclic boronate **6** by hydro-

[Q6] The use of this methodology to synthesize the beta-lactamase inhibitor QPX7728 provides a nice demonstration of its utility. However, the implication that this represents a 4-step synthesis from a commercially-available starting material, and the statement "compared with the previous synthesis, our concise and practical approach was cost-effective" ignores the fact that the starting material 3-bromo-4-fluoro-2-methoxybenzaldehyde costs an exorbitant amount (\$500,000/kg based on gram amounts). The wording should be revised to simply describe this as an efficient

alternative method; while it is important to acknowledge the published synthesis, cost comparisons should be avoided.

Response: We very much appreciate the reviewer's advice, the descriptions have been revised in the modified manuscript, and the reviewer can also find it as follows.

(±)-QPX7728 in 42% overall yield.⁴⁵ Compared with the previous synthesis,^{19,45} ~~our concise and practical approach was cost-effective, this as an efficient alternative method is~~ allowing for further optimization in the scalable production of QPX7728. ¶¶

[Q7] Also, as noted above, it would be useful to know if the boron insertion can be achieved with the carboxylate moiety already in place (presumably as an ester).

Response: Thanks for the reviewer's advice. During this research, we have planned to introduce an ester group in the positions ortho and para to the alkenyl group and tried to develop a concise route to synthesize QPX7728-Na, however, when the strongly electron-withdrawing substituent -CO₂R was introduced to the aromatic ring, the substrate become inert under the standard condition (presented in the following scheme). If the additive A5 was absent from the reaction system, only demethylated by-product was found without bicyclic boronate. Because both lone pair electrons and π electrons could coordinate with Lewis acid BBr₃, with 1.1 equivalent of BBr₃ under our standard conditions, we proposed that the lone pair electrons of oxygen atom competes with π electrons of alkene to form more stable complex A and inhibited the subsequent reaction. The reviewer can also find them as follows.

[Q8] • It is not at all clear how an MIC of 2.67 μ g/ml against E.coli is obtained for compound 3bp. It is generally accepted that the MIC corresponds to "no growth." According to the plot on page S31 of the Supporting Info, this is achieved at 2.2 log₁₀,

which would be well over 100 µg/ml. If it is the case that all the MICs are 100 µg/ml, then the antibacterial evaluation should be removed entirely from the manuscript since many organic compounds have modest antibacterial activity, in most cases by non-specific and uninteresting mechanisms of action. On the other hand, if the MIC vs. *E.coli* is in fact <3 µg/ml, it is indeed an interesting result that is worthy of reporting.

Response: Thanks for the reviewer's advice. The MIC of compound 3bp against *E.coli* was rechecked and the value is **xxx** 100 µg/ml, so the antibacterial evaluation have been removed entirely from the revised manuscript and SI. The reviewer can also find them as follows.

Preliminary biological evaluation. As we know, pan-β-lactamase inhibitors, such as taniborbactam,⁴⁷ VNRX-7145,⁹ and QPX7728,¹⁰ are currently under clinical development, and many bicyclic boronates are recognized for their potential application in antibacterial drug discovery.^{17,18} Thus, three selected products (**3ab**, **3bp**, and **16**) were evaluated for their preliminary antibacterial activities against gram-negative *Escherichia coli* (*E. coli*) and gram-positive *Staphylococcus aureus* (*S. aureus*), two most common pathogens causing healthcare-associated infections and bacteremia.⁴⁸ As illustrated in Fig. 7, all three compounds showed modest antibacterial activity, and **3bp** exhibited the best result against gram-negative bacterium *E. coli* with MIC in the low microgram range (2.67 ± 2.05 µg/mL), indicating that our novel bicyclic boronates could have potential use for further design and development of antibacterial agents.¶

Fig. 7 Preliminary screening of bicyclic boronates for preliminary antibacterial activities against *S. aureus* and *E. coli*.¶

Supplementary Inhibiting Activity on Escherichia coli (E. coli) and Staphylococcus aureus (S. aureus) in vitro

The microbial strains, gram-negative Escherichia coli (ATCC8739) and gram-positive Staphylococcus aureus (ATCC6538), were purchased from the Biobw biotechnology Co., Ltd., (Beijing, China). E. coli and S. aureus were inoculated into Luria-Bertani (LB) solid medium (tryptone 10 g/L, yeast extract 5 g/L, NaCl 10 g/L, and agar powder 15 g/L) with non-corresponding anti-biotics respectively, and incubated at 37 °C, until got Colony Forming Units. The normal appearing colonies of bacteria was taken into shake tube and incubated at 37 °C for 12 h to prepare a bacterial suspension, which was later used for inoculation.

The minimum inhibitory concentrations (MICs) of antimicrobial agents were determined by dilution methods²⁴. Compounds **3ab**, **3bp**, and **16** were dissolved in dimethyl sulfoxide (DMSO) to prepare stock solution, and then diluted into testing concentration, such as 400 µg/mL, 300 µg/mL, 225 µg/mL, 168.75 µg/mL, 126.56 µg/mL, 94.92 µg/mL, 71.19 µg/mL, 40.04 µg/mL, and 31.43 µg/mL. After that, 100 µL of the resulting solutions were added to 96-well plates and 100 µL of the diluted bacterial suspension ($V_{\text{liquid medium}}:V_{\text{bacterial suspension}}=5:1$) dispensed into the corresponding well, followed by overnight incubation at 37 °C for 12 h. At last, the optical density (OD) of the supernatant at 600 nm was recorded using the multimode microplate reader to calculate MIC₅₀. All tests were performed in triplicate.

Page Break

Reviewer #3:

Tian and co-workers report a new, transition metal-free synthetic protocol for bicyclic boronate synthesis via heteroatom-directed C(sp²)-H borylation. These compounds are relatively rare in the literature but are found in several important-lactamase inhibitors, as well as fluorescent sensors and functional materials. Hence, this report is likely to be of significant interest to the synthetic chemistry community, including medicinal chemists.

The scope of the method is fairly broad, and both electron-rich and electron-poor arene groups are compatible, as well as different alkene substitution patterns. [Q1] However, no Lewis basic heteroaromatic substrates (e.g., methoxy vinyl pyridines) are demonstrated, and one would assume that they would complex the BBr₃ non-productively? Notably, the reaction can be carried out cleanly on gram scale, which highlights its practicality. The authors also provide extensive examples of further derivatisations of the bicyclic boronates, which is great to see. Moreover, they applied their method to the synthesis of the ultrabroad-spectrum β -lactamase

inhibitor (\pm)-QPX7728, and also showed that several of the boronates displayed antibacterial activity against gram-negative bacterium *E. coli*.

This is a very well put together manuscript and was an enjoyable read. The Supporting Information is prepared to a good standard and the NMR spectra are all of high quality. Given the novelty of the approach to bicyclic boronates and the widespread interest in these unusual compounds, I have no hesitation in recommending this work for publication in Communications Chemistry.

Response to Reviewer 3

[Q1] However, no Lewis basic heteroaromatic substrates (e.g., methoxy vinyl pyridines) are demonstrated, and one would assume that they would complex the BBr_3 non-productively?

Response: Thanks for the reviewer's advice. The heteroaromatic substrate 4-methoxy-3-(prop-1-en-2-yl)pyridine was reinvestigated under our reaction conditions, as reviewer's assumed, no product was observed. In the revised manuscript, we have discussed the reactivity of BBr_3 was dramatically affected by pyridine (**A2**), 2,6-lutidine (**A3**), 2,6-di-*tert*-butylpyridine (**A5**) and the reasons in the section of "Mechanistic Considerations". The reviewer can also find them as follows.

Fig. 6 NMR experiments. A. ^{11}B NMR stacked spectra. B. ^1H NMR stacked spectra.

REVIEWERS' COMMENTS:

Reviewer #1 (Remarks to the Author):

All questions have been answered very well, so it is recommended to accept.

Reviewer #2 (Remarks to the Author):

The authors have provided a thorough response to my questions in the rebuttal letter. However, it is not sufficient to educate me regarding the negative results obtained with electron-withdrawing substituents and with directing groups NH₂ and SH. Negative results are just as important as positive results in defining the scope of a new reaction. These important negative results should be included in the manuscript.

Reviewer #3 (Remarks to the Author):

Having reviewed a previous version of the manuscript, I have no additional comments to add. I have read the authors' rebuttal letter in depth and they appear to have satisfactorily addressed all of the various comments, at least in my opinion. I am glad that they have opted to remove the MIC data, following Reviewer 2's feedback.

All in all I am happy to recommend the work for publication in Communications Chemistry.

General Response to Reviews

We appreciate the time and effort that the editor and reviewers dedicated to providing feedback on our previous manuscript (Ms. No **COMMSCHEM-22-0574B**) and are grateful for the insightful comments on and valuable improvements to our paper. We have carefully reviewed the comments and have revised the manuscript extensively. Our response is given in a point-to-point manner as follows.

Reviewer #2:

The authors have provided a thorough response to my questions in the rebuttal letter. However, it is not sufficient to educate me regarding the negative results obtained with electron-withdrawing substituents and with directing groups NH₂ and SH. Negative results are just as important as positive results in defining the scope of a new reaction. These important negative results should be included in the manuscript.

Response: Thanks for the reviewer's advice. The negative results have been added in the modified manuscript and the data of characterizations were added in the section of supplementary method and NMR spectra.

R² = CHO, **3fa**, n.d.

R² = CN, **3fb**, n.d.

3fc, n.d.

leased HBr. It is worth mentioning that when the substituent R² is an electron-withdrawing group such as -CHO (**3fa**), -CN (**3fb**), or a substrate with a heteroaromatic ring (**3fc**), the desired product cannot be detected under our standard conditions.¶

3fd, n.d.

3fe, n.d.

lographic analysis. It is also worth mentioning that no desired product can be detected when the substrate contains -NH₂ (**3fd**) or -SH (**3fe**) group under the standard conditions.¶

4-Methoxy-3-(prop-1-en-2-yl)benzaldehyde (1fa)

1fa

Following the general procedure on 5.0 mmol scale, the substrate **1fa** was obtained as colorless oil. Eluent: PE/EA (20:1), 176 mg, 20% yield; $^1\text{H NMR}$ (600 MHz, MeOD) δ 9.83 (s, 1H), 7.84 (d, $J = 8.5$ Hz, 1H), 7.70 (s, 1H), 7.14 (d, $J = 8.5$ Hz, 1H), 5.16 (s, 1H), 5.05 (s, 1H), 3.92 (s, 3H), 2.09 (s, 3H); $^{13}\text{C NMR}$ (151 MHz, MeOD) δ 191.60, 162.04, 143.26, 133.34, 131.45, 130.00, 129.61, 115.06, 110.73, 54.96, 21.76; **HRMS (ESI)**: Exact mass calcd for $\text{C}_{11}\text{H}_{13}\text{O}_2$ $[\text{M}+\text{H}]^+$: 177.0915, Found: 177.0912.

4-Methoxy-4-(prop-1-en-2-yl)benzoxazole (1fb)

1fb

Following the general procedure on 5.0 mmol scale, the substrate **1fb** was obtained as colorless oil. Eluent: PE/EA (20:1), 450 mg, 52% yield; $^1\text{H NMR}$ (400 MHz, CDCl_3) δ 7.54 (d, $J = 8.5$ Hz, 1H), 7.45 (s, 1H), 6.91 (d, $J = 8.6$ Hz, 1H), 5.22 – 5.17 (m, 1H), 5.09 – 5.04 (m, 1H), 3.88 (s, 3H), 2.07 (s, 3H); $^{13}\text{C NMR}$ (101 MHz, CDCl_3) δ 160.13, 142.40, 134.09, 133.11, 133.09, 119.31, 116.82, 111.24, 103.95, 55.83, 22.81; **HRMS (ESI)**: Exact mass calcd for $\text{C}_{11}\text{H}_{15}\text{N}_2\text{O}$ $[\text{M}+\text{NH}_4]^+$: 191.1185, Found: 191.1180.

4-Methoxy-3-(prop-1-en-2-yl)pyridine (1fc)

1fc

Following the general procedure on 5.0 mmol scale, the substrate **1fc** was obtained as colorless oil. Eluent: PE/EA (10:1), 470 mg, 63% yield; $^1\text{H NMR}$ (300 MHz, CDCl_3) δ 8.38 (d, $J = 5.7$ Hz, 1H), 8.29 (s, 1H), 6.76 (d, $J = 5.7$ Hz, 1H), 5.18 (s, 1H), 5.09 (s, 1H), 3.86 (s, 3H), 2.08 (s, 3H); $^{13}\text{C NMR}$ (101 MHz, CDCl_3) δ 162.87, 150.49, 149.59, 140.71, 128.45, 116.63, 106.18, 55.32, 22.91; **HRMS (ESI)**: Exact mass calcd for $\text{C}_{12}\text{H}_{16}\text{NO}$ $[\text{M}+\text{C}_3\text{H}_5]^+$: 190.1227, Found: 190.1229.

2-(prop-1-en-2-yl)Aniline (1fd)

1fd

Following the general procedure on 5.0 mmol scale, the substrate **1fd** was obtained as colorless oil. Eluent: PE/EA (5:1), 386 mg, 58% yield; $^1\text{H NMR}$ (600 MHz, CDCl_3) δ 7.09 (m, 2H), 6.77 (td, $J = 7.5, 1.1$ Hz, 1H), 6.73 (dd, $J = 7.9, 1.0$ Hz, 1H), 5.33 (m, 1H), 5.12 – 5.08 (m, 1H), 3.82 (s, 2H), 2.11 (s, 3H); $^{13}\text{C NMR}$ (151 MHz, CDCl_3) δ 143.55, 142.90, 129.33, 128.30, 127.98, 118.28, 115.64, 115.39, 23.98; **HRMS (ESI)**: Exact mass calcd for $\text{C}_9\text{H}_{12}\text{N}$ $[\text{M}+\text{H}]^+$: 134.0969, Found: 134.0970.

Scheme: 1

O-(2-(prop-1-en-2-yl)phenyl) dimethylcarbamothioate (**18**)

To a mixture of **1bb** (268 mg, 2.0 mmol) in DMF (10 mL) was added NaH (120 mg, 60% in mineral oil, 3.0 mmol, 1.5 equiv) at 0 °C under N₂ atmosphere. After 30 min *N,N*-Dimethylcarbamothioic chloride was added and the reaction mixture was then stirred at room temperature for 18 h. The reaction was quenched by saturated NH₄Cl and then transferred to a separatory funnel, and extracted with ethyl acetate (EA, 30 mL×3). The combined organic layers were washed with brine, dried with Na₂SO₄, filtered, and concentrated in vacuo. The crude mixture was purified by silica gel column chromatography to afford the desired product **18** as pale yellow oil in 85% yield (374 mg).

Eluent: PE/EA (10:1), ¹H NMR (600 MHz, CDCl₃) δ 7.30 – 7.26 (m, 2H), 7.24 – 7.21 (m, 1H), 7.05 (d, *J* = 8.3 Hz, 1H), 5.17 – 5.16 (m, 1H), 5.06 – 5.05 (m, 1H), 3.44 (s, 3H), 3.31 (s, 3H), 2.07 (s, 3H).; ¹³C NMR (151 MHz, CDCl₃) δ 187.6, 150.6, 141.3, 136.7, 129.0, 127.6, 125.9, 124.0, 116.0, 43.2, 38.6, 23.7.; HRMS (ESI): Exact mass calcd for C₁₂H₁₅NOS[M+Na]⁺: 244.0767, Found: 244.0754.

S-(2-(prop-1-en-2-yl)phenyl) dimethylcarbamothioate (**19**)

To a 25-mL sealed Schlenk tube containing a magnetic stir bar was added **18** (354 mg, 1.6 mmol), 1-butyl-3-methylimidazolium tetrafluoroborate (ion liquid, 0.8 mL) and diphenyl ether (1 mL) under nitrogen atmosphere. The reaction mixture was then heated to 240 °C and stirred for 4 hours. After that, the resulting solution was cooled to room temperature and mixture was directly purified by column chromatography to afford the desired product **19** as yellow oil in 76% yield (270 mg).

Eluent: PE/EA (5:1), ¹H NMR (600 MHz, CDCl₃) δ 7.51 (dd, *J* = 7.8, 1.5 Hz, 1H), 7.33 (td, *J* = 7.5, 1.4 Hz, 1H), 7.26 (td, *J* = 7.6, 1.6 Hz, 1H), 7.22 (dd, *J* = 7.6, 1.7 Hz, 1H), 5.16 (p, *J* = 1.6 Hz, 1H), 4.88 (dd, *J* = 2.1, 1.0 Hz, 1H), 3.01 (d, *J* = 39.0 Hz, 6H), 2.09 – 2.03 (m, 3H).; ¹³C NMR (151 MHz, CDCl₃) δ 166.8, 148.7, 145.2, 137.4, 129.1, 128.5, 127.2,

126.0, 115.5, 36.7, 24.2; **HRMS (ESI)**: Exact mass calcd for $C_{12}H_{15}NOS[M+H]^+$: 222.0947, Found: 222.0956.

2-(prop-1-en-2-yl)benzenethiol (**1fe**)

To a one-necked 50-mL round bottomed flask equipped with a Teflon-coated magnetic stir bar was added **19** (50 mg 0.23 mmol, 1.0 equiv) and dry ether (10 mL) under nitrogen atmosphere in an ice bath. Lithium aluminum hydride ($LiAlH_4$, 17 mg, 0.46 mmol, 2 equiv) was then added in portion and the reaction mixture was stirred for 30 min. The excess $LiAlH_4$ is quenched by the slow addition of water (10 mL). To the resulting mixture was added 1 N HCl (2 mL) and then transferred to a separatory funnel, and extracted with ether (10 mL \times 3). The combined organic layers were washed with brine, dried with Na_2SO_4 , filtered, and concentrated in vacuo. The crude mixture was purified by silica gel column chromatography to afford the desired product **1fe** as colorless oil in 31% yield (11 mg).

Eluent: PE/EA (20:1), **1H NMR** (600 MHz, $CDCl_3$) δ 7.60 (dd, $J = 7.8, 1.4$ Hz, 1H), 7.21– 7.18 (m, 1H), 7.17– 7.15 (m, 1H), 7.12 – 7.11(m, 1H), 5.33 – 5.32 (m, 1H), 5.02 – 5.01 (m, 1H), 2.12 – 2.12 (m, 3H).; **^{13}C NMR** (600 MHz, $CDCl_3$) δ 143.6, 142.9, 134.1, 128.1, 127.8, 126.7, 126.3, 117.4, 24.2. **HRMS (ESI)**: Exact mass calcd for $C_9H_{10}S[M]^+$: 150.0498, Found: 150.0503.

03-94.20.fid

03-94.22.fid